# Rational Design of Pectin–Chitosan Polyelectrolyte Nanoparticles for Enhanced Temozolomide Delivery in Brain Tumor Therapy

**DOI:** 10.3390/biomedicines12071393

**Published:** 2024-06-23

**Authors:** Vladimir E. Silant’ev, Andrei S. Belousov, Fedor O. Trukhin, Nadezhda E. Struppul, Mikhail E. Shmelev, Aleksandra A. Patlay, Roman A. Shatilov, Vadim V. Kumeiko

**Affiliations:** 1School of Medicine and Life Sciences, Far Eastern Federal University, Vladivostok 690922, Russia; belousov.ands@gmail.com (A.S.B.); trukhin.fo@dvfu.ru (F.O.T.); struppul.nye@dvfu.ru (N.E.S.); shmelev.m.e@gmail.com (M.E.S.); patlay.al@mail.ru (A.A.P.); shatilov.ra@dvfu.ru (R.A.S.); 2Laboratory of Electrochemical Processes, Institute of Chemistry, Far Eastern Branch of Russian Academy of Sciences, Vladivostok 690022, Russia; 3A.V. Zhirmunsky National Scientific Center of Marine Biology, Far Eastern Branch of Russian Academy of Sciences, Vladivostok 690041, Russia

**Keywords:** nanocarrier, drug delivery systems, carbohydrates, polysaccharides, biomaterials, viscoelastic properties, blood–brain barrier, brain tumors, temozolomide

## Abstract

Conventional chemotherapeutic approaches currently used for brain tumor treatment have low efficiency in targeted drug delivery and often have non-target toxicity. Development of stable and effective drug delivery vehicles for the most incurable diseases is one of the urgent biomedical challenges. We have developed polymer nanoparticles (NPs) with improved temozolomide (TMZ) delivery for promising brain tumor therapy, performing a rational design of polyelectrolyte complexes of oppositely charged polysaccharides of cationic chitosan and anionic pectin. The NPs’ diameter (30 to 330 nm) and zeta-potential (−29 to 73 mV) varied according to the initial mass ratios of the biopolymers. The evaluation of nanomechanical parameters of native NPs demonstrated changes in Young’s modulus from 58 to 234 kPa and adhesion from −0.3 to −3.57 pN. Possible mechanisms of NPs’ formation preliminary based on ionic interactions between ionogenic functional groups were proposed by IR spectroscopy and dynamic rheology. The study of the parameters and kinetics of TMZ sorption made it possible to identify compounds that most effectively immobilize and release the active substance in model liquids that simulate the internal environment of the body. A polyelectrolyte carrier based on an equal ratio of pectin–chitosan (0.1% by weight) was selected as the most effective for the delivery of TMZ among a series of obtained NPs, which indicates a promising approach to the treatment of brain tumors.

## 1. Introduction

Cancer is the second leading cause of death followed by cardiovascular disease [1]. There were 20 million new cases of cancer worldwide in 2022, but this percentage is predicted to increase by 77% to 35 million by 2050 [2]. Targeted delivery of drugs to malignant tumors is one of the most pressing problems in cancer therapy. It can reduce the negative impact on healthy cells during chemotherapy and increase selectivity against tumor cells [3]. Additional challenges arise when treating tumors in hard-to-reach areas. For example, drug delivery to brain tissue via the bloodstream faces the problem of drug transport across the blood–brain barrier (BBB). An alternative option is a direct injection of the drug into the tumor site or through the implantation of a material preloaded with the drug. This poses a risk of side effects that can lead to the death of the patient in case of brain damage [4].

An optimal effect depends on the ability to maintain drug concentrations at the target site for a sufficient time while avoiding systemic toxicity. This requirement is almost never achieved using standard chemotherapeutic agents. For example, most small-molecular-weight drugs are rapidly eliminated by hepatic metabolism and renal excretion, so only a limited portion of the drug reaches the brain tumor [5].

A chemotherapy drug may be toxic to non-target organs and cause systemic toxicity. The prospects for using a drug are determined by its ability to achieve clinically useful concentrations in the target organ or blood plasma. However, even after achieving adequate plasma drug concentrations, drugs must overcome physiological barriers to reach the extracellular space of the tumor. A drug that is poorly distributed in the brain can be loaded into a nanocarrier system that interacts with the microvascular endothelium at the level of the BBB. This should lead to an increase in the concentration of the drug in the brain parenchyma [6].

Progress in the field of nanotechnology has made it possible to create new colloidal forms that can effectively adsorb drugs and change their pharmacological and biopharmaceutical characteristics. The unique physicochemical and technological attributes of nanomaterial-based therapy have enabled its successful utilization in cancer treatment [7]. Nanoscale polymer systems loaded with chemotherapy show enhancing the efficacy of diverse tumor treatment approaches [7,8]. Although nanotechnology strategies for diagnosing and treating cancer have advanced, progress in effectively treating brain tumors has been limited. A major obstacle persists in enabling therapeutic agents to breach the BBB and attain precise targeting capabilities. Numerous invasive and non-invasive methods, along with a range of nanocarriers and their modifications, are under extensive investigation for brain tumor therapy [9,10,11]. However, significant clinical applications have not been achieved. Chemotherapy and surgery continue to be the primary treatment modalities for brain tumors [12].

Drug carriers can also be delivered through the BBB by cellular transport. They deliver particles through the BBB to the target location. Nanoparticles (NPs) that are elaborated to cross the BBB may be polymers and contain poly(lactic-co-glycolic acid) and polylactic acid, or may be based on liposomes, gold, silver, and zinc oxide [13]. In this regard, the development of micro- and nanosystems for drug delivery is one of the most urgent modern scientific topics [14]. Nanoscale delivery systems can have specific drug release behavior that increases the drug concentration at the target site and decreases the drug concentration at the non-target site, thereby reducing side effects. It is easy to implement combination treatments to achieve a synergistic effect by using nanoscale delivery systems. Thus, NPs provide an excellent investigation platform on drugs targeting brain tumors [15]. There are numerous papers devoted to the synthesis and testing of particles used for the above purpose. Among polymer constructs, we can emphasize the biological substances isolated from natural sources or obtained by their chemical transformation. These materials could be used for the nanoscale carrier design and synthesis that provide specific effects on signal transduction pathways [16].

Various polysaccharides and their derivatives are the main components of the extracellular matrix of the human brain [17]. Therefore, biomaterials based on them may be more convenient for the development of new therapeutic agents. The main advantages of polysaccharides are low toxicity, biodegradability, biocompatibility, and minimal immunogenicity. These properties, combined with simple chemical modification and low raw material costs, allow for a large-scale production of therapeutic systems for oncology. Biopolymers are used to create various forms of materials such as particles, films, and hydrogels [18,19]. Polysaccharides can also be specifically designed to deliver drugs and bioactive agents directly to tumor cells. This may help to reduce drug side effects and improve treatment efficacy [20]. When it comes to brain tumors, polysaccharide NPs are potentially suitable for crossing the BBB because they stimulate various mechanisms of transporting substances into the cell [21].

Materials based on two polysaccharides, cationic chitosan and anionic pectin, were used in this work. Chitosan is a derivative of the second most abundant polysaccharide in nature, chitin. It is the only cationic polyelectrolyte of a natural origin (Figure 1a). Pectin is a natural polymer, one of the main components of a plant cell wall. The structure of pectin is based on α-1,4-galacturonic acid residues partially esterified with methoxyl groups (Figure 1b). These groups can be chemically modified by alkaline de-esterification to produce a range of biopolymers characterized by different physicochemical properties.

Pectin-based materials have been proposed for targeted drug delivery, wound healing, and tissue engineering in biomedicine. Pectins can be readily used to produce films, microparticles, microcapsules, and fibers. It is worth noting that the synthesis of pectin NPs involves many challenges, as evidenced by the limited number of works. At the same time, pectin-based biomaterials can mimic polysaccharides of the extracellular matrix of the mammalian brain. Pectin contributes to the creation of a favorable microenvironment for nerve cells, simulates cell migration, and affects the processes of tissue repair [22,23]. Moreover, chemical modifications of natural pectin or the improvement in functional additives offer great opportunities for the development of pectin-based biomaterials for various biomedical applications.

The NPs’ charge influences the interaction between nanomaterials and cells [24]. Charged forms of NPs have the following advantages compared to neutral ones: (1) High stability—due to the lack of electrostatic interaction, neutral NPs have low physical stability and cannot inhibit the self-aggregation. Meanwhile, surface charges prevent the polymerization and flocculation of NPs through electrostatic interaction. (2) High permeability—NPs can interact with cells through surface electrostatic charges, improving drug accumulation in cells [25]. For example, cationic NPs interact with the negatively charged BBB and are transported into the brain [26]. However, cationic NPs in particular are strongly attracted by negatively charged phospholipid residues and a few proteins, and the positive surface charge leads to systemic side effects [27].

Polyelectrolyte complexes (PECs) based on the two oppositely charged polyelectrolytes chitosan and pectin were obtained by a modified ionic gelation method. Adjusting the number of ionic functional groups capable of gelation and the stoichiometric ratio of oppositely charged groups makes it possible to create biocompatible nanocarriers of therapeutic agents. By fine-tuning, polyelectrolyte complexes of different sizes, viscoelastic properties, and surface charge can be obtained. The difference should affect the drug loading and release processes.

## 2. Materials and Methods

### 2.1. Materials

High-molecular chitosan (molecular mass (Mm): 200 kDa, degree of deacetylation (DD): ~80%, JSC “Bioprogress”, Moscow, Shchelkovo), glacial acetic acid (VECTON, Saint Petersburg, Russia, 98%), pectin (Mm: 120 kDa, degree of esterification (DE): ~55%, Herbstreith & Fox GmbH & Co Group, Neuenbürg, Germany), poly-L-lysine 0.1% (Sigma Aldrich, Taufkirchen, Germany), formvar (Sigma Aldrich, Taufkirchen, Germany), chloroform (Saint Petersburg, VECTON, Russia, 99%, MF), mica (Mica Power Co., Ltd., Dongguan, China), bovine serum albumin (BSA) (Fisher BioReagents^TM^, Pittsburgh, PA, USA), tetrazolium dye MTT (Sigma Aldrich, Taufkirchen, Germany), urea (Molecule Ltd., London, UK, 99% of the total), 2-[4-(2-hydroxyethyl)piperazin-1-yl]ethane-1-sulfonic acid (HEPES) (Suzhou Yacoo Science Co., Suzhou Industrial Park, Suzhou, China, 99.5% MF), D-glucose (Amresco Ink, Solon, OH, USA, 99% MF), sodium chloride (Sigma Aldrich, Taufkirchen, Germany, 99.5% MF), calcium chloride dihydrate (Amresco Ink, Solon, OH, USA, 99% MF), magnesium chloride hexahydrate (Sigma Aldrich, Taufkirchen, Germany, 99.5% MF), sodium bicarbonate (Amresco Ink, Solon, OH, USA, 99% MF), temozolomide (TMZ) (Temomid, lyophilizate for preparation of solution for infusion, Jodas Expoim Pvt. Ltd., Telangana, India), sodium hydroxide (Sigma Aldrich, Taufkirchen, Germany, 99.5% MF), copper sulfate (Sigma Aldrich, Taufkirchen, Germany, 99.5% MF), dimethyl sulfoxide (DMSO) (99% purity, Dimexide, Usolye-Siberian Chemical Pharmaceutical Plant JSC, Irkutsk region, Usole-Sibirskoye, Russia), potassium hydrophosphate (Molecule Ltd., London, UK, 99% SCH), potassium dihydrogen phosphate (Amresco Ink, Solon, OH, USA, 99% SCH), and Tween-20 (Suzhou Yacoo Science Co., Suzhou Industrial Park, China, 99.5% SCH) were used. Deionized water used in the experiments was obtained by using the Milli-Q IQ-7000 water purification system (Merck Millipore, Rahway, NJ, USA).

### 2.2. Obtaining of Low-Esterified Pectin

Low-esterified pectin was obtained from commercial apple pectin by using the acid method [28]. DE was determined by the titrimetric method according to the National State Standard [29].

### 2.3. Preparation of NPs

The chitosan solution with a concentration of 1.5 wt.% was prepared by dissolving the initial polysaccharide powder in 2 vol.% glacial acetic acid. The solution was evenly mixed on a magnetic stirrer at 800 rpm for 4.5 h, similar to the method from the article [30]. The suspension was centrifuged for 15 min at a speed of 5000 g at room temperature to remove the undissolved chitosan. The powder of modified pectins with a DE of 10% was suspended in deionized water at a concentration of 1 wt.% and heated in a water bath at a constant stirring for 1 h at 60 °C. The pectin solution was clarified by centrifugation at 3000× *g* for 40 min. Stock solutions of both biopolymers were brought with deionized water to the desired concentration of each polyelectrolyte in the range from 0.01 wt.% to 0.5 wt.%.

NPs based on polyelectrolytes were prepared by the ion gelling technique. A pectin solution of 1 mL was added into a 10 mL vial and stirred at 600 rpm. A chitosan solution of 1 mL was added drop by drop at a rate of 1 drop per 10 s using a dispenser located 10 cm above the surface of the solution with a nozzle hole diameter of 1.125 μm. The stirring speed was increased to 800 rpm as the volume increased gradually and mixed for 40 min. The suspension was treated twice with ultrasound (US) (Ultrasonic Cleaner Manufacturer Expert, Beijing, China) at a frequency of 70 kHz and a temperature of 30 °C for 5 min to increase stability. NPs were dispersed at a speed of 800 rpm for 1 min between approaches. Final solutions were mixed at 600 rpm for 18 h to complete the gelation process.

Suspensions of NPs were washed by centrifugation (Eppendorf Centrifuge 5910 Ri, Hamburg, Germany). An aliquot of a suspension of 950 µL was previously added to 2 microprobes per 2 mL. The dispersion was centrifuged for 5 min at 22,132× *g*. A supernatant of 640 μL was taken from the resulting solutions of each microprobe and the same amount of deionized water was added. The suspensions were dispersed 10–15 times. The dispersions were centrifuged and superadded, and deionized water was added and dispersed at the same parameters 2 more times. The samples were put in the refrigerator for 4 °C after all treatments.

### 2.4. Preparation of Model Buffer Systems and Model Drugs

Buffer solutions’ model blood plasma fluid (MBPF) (90 g of water, 7.5 g of BSA, 0.018 g of urea, and 0.29 g of HEPES, similar in polarity and Mm to lipids, while not shifting pH due to the absence of ions, 0.072 g of D-glucose, 0.322 g of sodium bicarbonate, 0.019 g of potassium hydrophosphate, 0.01 g of calcium chloride dihydrate, 0.002 g of magnesium chloride hexahydrate, and 0.343 g of sodium chloride) [31,32] and model cerebrospinal fluid (MCF) (95 g of water, 0.25 g of BSA, 0.048 g of D-glucose, 0.322 g of sodium carbonate, 0.0088 g of potassium chloride, 0.0038 g of potassium hydrophosphate, 0.005 g of calcium, 0.43 g of sodium chloride, 0.003 g of magnesium chloride) [33] were prepared with pH 7.4 and 7.3, respectively. TMZ (MM: 194.151 g/mol) was dissolved in DMSO up to 0.388 mM for sorption studies and up to 50 mM for rheology testing. The resulting solution was mixed for 1 day at 25 °C until the chemotherapy drug was completely dissolved. The completely dissolved dosage form was taken to be one in which no precipitation was observed during centrifugation at 10,000× *g*.

### 2.5. Transmission Electron Microscopy (TEM)

Structural features of the NPs were determined by the TEM method using the Libra 120 microscope (Carl Zeiss, Oberkochen, Germany). The NP samples were pre-treated with US before imaging. Then, a drop of the suspension was placed on a copper TEM grid pre-coated with a formvar polymer dissolved in chloroform. The samples were incubated at room temperature for 15 min, excess solvent was removed, and the samples were placed in the vacuum chamber of the microscope for examination.

### 2.6. Atomic Force Microscopy (AFM)

Research was carried out on a high-resolution AFM Bioscope resolve (Bruker, Billerica, MA, USA). NPs were deposited on a freshly cleaved mica sheet coated with poly-L-lysine, which was attached to a 35 mm Petri dish. A poly-L-lysine coating was created by the following procedure: a drop of a 0.01% poly-L-lysine solution was applied to mica and left for 5 min, then removed by a pipette and rinsed with deionized water.

NPs’ morphology and mechanical properties were examined in semi-contact mode PeakForce Quantitative nanomechanical mapping (PF-QNM) using an SNL-A cantilever (Bruker Optics GmbH & Co., Ettlingen, Germany) in a DMEM cell culture medium (Capricorn, Düsseldorf, Germany).

Prior to each AFM assay, the cantilever deflection sensitivity and spring constant were estimated by the thermal noise method under the scanning media, and several force curves from a hard sample (sapphire) were created. The tip radius was estimated by performing the scanning of a Ti rough sample and calculated using Bruker Nanoscope 9.2 analysis software. The Poisson’s ratio was set to 0.5. All samples were analyzed using Bruker Nanoscope analysis software (version 1.40) and open-source Gwyddion software (version 2.62). To calculate the values of adhesion and the modulus of elasticity (Young’s modulus), data from samples obtained in liquid were used. Images for studying the diameter of NPs were processed in Image J 1.53t software.

### 2.7. Rheology

Rheological measurements are based on the deformation of a body under the impact of an external shear stress τ, which is applied along a tangent to its upper surface, while the lower one is fixed on a fixed plane. Measurements were performed on a Haake Mars 40 rotary rheometer (Thermo Scientific, Karlsruhe, Germany). Torque was varied in the range of 5 × 10^−8^ to 0.2 N⋅m and in the frequency range of 10^−5^–100 Hz (results are given for the most reliable range of 10^−2^–10 Hz). The normal component of the force was determined to an accuracy of 0.001 N. The cone-plane cell type was used. The cone parameters were a 1° angle and 20 mm diameter (cylinder rotor C201/Ti and lower plate TMP20, Thermo Scientific, Karlsruhe, Germany). The volume of the test samples at each measurement was 50 μL. The deformation of the material in both rheometers was carried out according to the Searle principle: the lower part of the rheometer cell remained stationary during operation, while the upper cone rotated.

### 2.8. Fourier Transform Infrared (FT-IR) Spectroscopy

Measurements were carried out on Shimadzu IRAffinity-1S (Kyoto, Japan) with a prefix for the spectroscopy of disturbed total reflection PIKE technologies MIRacle™ 10 (Madison WI, USA). The fixation of the samples on the plate was carried out by pressing a flat tip. The spectra were taken in the range of 400–4000 cm^−1^ with a resolution of 4 cm^−1^. They were also obtained in the absorbent mode with Happ–Genzel apodization.

### 2.9. Zeta-Potential

The magnitude of the zeta-potential and the size were estimated by photon correlation spectroscopy on the Zetasizer Nano ZS analyzer (Malvern Instruments, Worcestershire, UK). Sample preparation involved US processing of samples into a Sonopuls HD 3200 homogenizer (Bandelin, Berlin, Germany) with a frequency of 20 kHz. The US treatment time was 3 min. Suspensions of 1 mL were placed in a DTS0012 cuvette and installed in a thermostable cuvette compartment of the analyzer after US treatment.

### 2.10. NPs’ Stability

The stability of NPs was studied in an MBPF. The solution was prepared by diluting HBSS 25 times (3.6 g buffer in 86.4 g water) and adding 0.004 g glucose, 7 g BSA, 0.018 g urea, and 0.23 g HEPES [34]. The sample was stored with stirring at 200 rpm at a temperature of 37 °C, simulating the conditions of human blood. The qualitative composition of the suspension was confirmed by AFM during the first 3 days.

### 2.11. Immobilization and Release of Drugs

The suspension of PEC NPs after ultrasound treatment was mixed with the TMZ solution in a volume ratio of 2:1 at 25 °C.

To quantify the amount the loaded TMZ, an indirect method was used. To carry out this, at the centrifuge stage of NPs’ preparation, the supernatant was separated and analyzed. TMZ concentration was evaluated by a UV–Vis spectrophotometer using a calibration curve. Measurements were carried out on a SHIMADZU UV-Vis spectrophotometer (λ = 330 nm, extinction coefficient: E^330^ = 6700 M^−1^ cm^−1^ and IMPLEN NanoPhotometer N50 (Implen GmbH, München, Germany) at 25 °C and/or 37 °C with a built-in digital dry bath. Samples were placed in 1 mm and 10 mm path length quartz cuvettes, respectively. TMZ immobilization (loading) efficiency and TMZ immobilization capacity were calculated according to the next equations:$$TMZ\;immobilization\;efficiency=\frac{C_{init}-C_{supernat}}{C_{init}}\cdot 100\%$$
$$TMZ\;immobilization\;capacity=\frac{C_{init}-C_{supernat}}{C_{NPs}}\cdot 100\%$$
where $C_{init}$—initial amount of TMZ;

$C_{supernat}$—amount of TMZ in supernatant;

$C_{NPs}$—amount of NPs.

The sorption characteristics of NPs were investigated as a function of pH solution changes, adsorbent composition (change in weight content of chitosan toward pectin), and contact time in separate aqueous and physiological solutions at constant temperature. The linear correlation coefficients of Freundlich isotherms and Brunauer–Emmett–Teller (BET) isotherms were obtained.

The linear form of the Freundlich equation was used,
$$log(a)=logK_{F}+logC_{b},$$
and the linear form of the BET model,
$$\frac{C_{b}}{a\cdot (C_{s}-C_{b})}=\frac{1}{a_{m}\cdot K_{BET}}+\frac{(K_{BET}-1)\cdot C_{b}}{a_{m}\cdot K_{BET}\cdot C_{s}}.$$

### 2.12. Statistical Analysis

All experiments, unless differently specified, were performed in triplicate, and quantitative data were shown as the mean ± standard deviation after an analysis of variance (ANOVA), which enabled the comparison of the data from the study groups via GraphPad Prism 8 software (GraphPad Software Inc., San Diego, CA, USA). Statistical differences were designated as significant if the *p*-values were less than 0.05 (* *p* ≤ 0.05) and highly significant if the *p*-values were less than 0.01 (** *p* ≤ 0.01), less than 0.001 (*** *p* ≤ 0.001), or less than 0.0001 (**** *p* ≤ 0.0001). The Mann–Whitney U-test was used to assess the differences between two independent samples.

## 3. Results

### 3.1. Phase Behavior, Morphology, and Topography of NPs

Different structures were obtained at different ratios—NPs, microparticles, aggregates, or nothing visible turned up. The dependence of the possibility of obtaining NPs on the concentration of polymers and their ratio is shown in the phase diagram in Figure 2a. The areas in green and red colors refer to blends in which phase separation occurs. The concentration of both polyelectrolytes occurs in one of the phases. This is due to their association through the formation of PECs. Phase behavior of this type is referred to as the associative type. The area inside the green region shows PEC compositions close to stoichiometric. In this case, homogeneous suspensions are formed. Outside this region are heterogeneous (red region) or soluble, differing from stoichiometric.

It was found by high-resolution TEM (Figure 2b) that the average diameter of the nanoscale particles increased with increasing chitosan content in the initial mixtures from 30 to 300 nm. In addition, clearly separated dense inner spheres with a less dense shell, possibly due to the excess of the cationic polyelectrolyte, appear in the NPs’ structure.

AFM images of NPs with a mass ratio of pectin to chitosan of 0.1:0.1, 0.1:0.3, and 0.1:0.5, taking into account the physicochemical properties, are shown in Figure 3a. Young’s modulus and adhesion were measured using AFM and evaluated in the Nanoscope 9.2 analysis program. The samples were imaged in a culture medium and their sizes were as close as possible to the originally obtained NPs in comparison with the TEM method. The average diameters of the NPs increased several times because of this reason. The increase in particle size from software processing results may be due to an insufficient resolution of nanoscale objects in close proximity to each other. Additionally, the stability of NPs in the MBPF model environment was evaluated. The studies were performed by AFM for three days daily for all formulations marked in green in Figure 2a. It is noted that most of the NPs maintained their stability. A slight segregation was observed for samples with high chitosan content in the initial mixtures.

Average dimensions of NPs were calculated in the Image J 1.53t program from AFM images. In this software, an area value from 30 to 1000 nm was taken. The dependence of the diameter of NPs on their composition is shown in Table 1. Unfortunately, the size dependence on composition does not correlate with the TEM data for all samples. This can be explained by the specific features of the software. Nevertheless, the maximum particle size was not more than 394.5 (IQR 81.8), which can be attributed to a positive practical result. The surface charge was characterized by the values of the zeta-potential. The mean values for the original chitosan and pectin in the dissolved state were 34.1 mV and −39.9 mV, respectively. Zeta-potential for PECs varies from −28.4 mV to 73.4 mV with an increase in chitosan concentration in the compositions. After increasing the mass fraction of chitosan to 0.5 wt.%, the zeta-potential is set in the range of 55 to 73.4 mV. Based on article [35], zeta-potential values close to and exceeding 30 mV suggest high stability of NPs. The dynamic light scattering data in almost all cases showed significantly larger dimensions than those obtained by AFM and TEM. This result is probably due to insufficiently effective methods for these types of materials. In addition, similar problems were observed when using laser NP analyzers from other manufacturers. The dependence of the zeta-potential and polydispersity index (PDI) on their mass ratio is shown in Table 1. PDI (dimensionless value) varies in a rather large range from 0.182 to 0.530. The index value is very sensitive to the presence of contaminants that affect the monodispersity of NPs. Such a spread possibly means an insufficiently efficient separation of the suspension from unreacted molecules. Nevertheless, the PDI values do not exceed 0.7, which is suitable for the dynamic light scattering method [36].

### 3.2. Viscoelastic Properties of NPs

NPs’ composition determined a significant impact on their mechanical properties. PECs, prepared of 0.2 wt.% chitosan and 0.2 wt.% pectin, showed the highest value of Young’s modulus, a median of 394.5 kPa (IQR 163.5 kPa) (Figure 3b,d). Chitosan NPs with lower pectin concentration were the softest. Young’s modulus of the NPs containing 0.4 wt.% chitosan and 0.1 wt.% pectin was 58.16 kPa (IQR 18.36 kPa). No significant differences in NPs’ mechanical properties were observed between the NPs with the low pectin concentration (Figure 3). Atomic force microscopy showed NPs’ mechanical stability in cell culture media, whose composition is close to actual human blood (Figure 3b,d). No significant effect on NPs’ adhesion of PEC composition was found (Figure 3c,e).

### 3.3. FT-IR Spectroscopy of NPs

Structural features of the obtained colloidal materials were investigated by IR spectroscopy. Figure 4 shows the spectra of the initial powders of pectin (DE: 10%) and chitosan (DD: 80%), as well as suspensions NP-01-01 and NP-01-05 and material based on pectin of 0.5 wt.% and chitosan of 0.1 wt.%. Band assignment shown in the spectrum is made for the chitosan according to [37,38] and pectin according to [39,40].

The vibrational spectrum of the chitosan powder (Figure 4(1)) is represented by bands at 3361 and 3436 cm^−1^, characteristic of the hydrogen and hydroxyl interactions in the vibrational region. Similar to the spectrum of pectin, chitosan exhibits a broad peak at 2874 cm^−1^, which corresponds to the symmetric and asymmetric vibrations of v_s_(C-H) and v_as_(C-H) bonds. A double peak at 1379 cm^−1^ and 1422 cm^−1^ is observed in the region of δ(C-H) deformation vibrations similar to the double peak of the pectin spectrum. Vibrations of the pyranose ring structure v(C-O) band are shown at 1075 cm^−1^. Peaks of the chitosan spectrum at 1591 and 1653 cm^−1^ correspond to fluctuations of δ(N-H) amine and v(C=O) acetylamide groups. Other vibrations of the N-H amino groups are at 3292 cm^−1^ and a number of peaks in the fingerprint region.

All informative regions of polyelectrolyte vibrations were considered in this study. The peak corresponding to the v(O-H) bonding region of water and hydroxyl groups of macromolecules, as well as vibrations corresponding to hydrogen interactions, appears on the spectrum of initial pectin (Figure 4(2)) at 3329 cm^−1^. The bands at 2936 cm^−1^ and 2879 cm^−1^ correspond to symmetric and asymmetric vibrations of the hydrocarbon bonds v_s_(C-H) and v_as_(C-H), respectively. The peak at 1407 cm^−1^ corresponds to the deformation vibrations of δ(C-H), and 1089 cm^−1^ corresponds to the v(C-O) bond vibration regions of the monomeric units of macromolecules. The bands on the spectrum of pectin at 1729 and 1597 cm^−1^ correspond to v(C=O) vibrations of esterified carboxyl groups and their charged form, respectively.

Most of the peaks characterizing both polysaccharides retain their positions on the spectra of polyelectrolyte complexes (Figure 4(3–5)). It should be noted that the final positions and intensities of the bands in the spectra of NPs and heterogeneous insoluble systems do not practically differ from each other. A new band appears at ~955 cm^−1^, which, based on literature data, most likely refers to an additional fluctuation of the glycosidic bond of the polysaccharide structure. The band at ~1530 cm^−1^ corresponds to the protonated form of the amino group (-NH_3_^+^) of chitosan appearing during PEC formation (Table 2). The band of the protonated form of the carboxyl groups of pectin retains its position and intensity is almost unchanged. A possible variant of interactions between polyelectrolytes may be ionic interactions supported by multiple hydrogen bonds.

### 3.4. Rheological Properties of NP Suspensions

Rheological properties of the obtained suspensions were characterized by means of two modes: oscillation and yield stress determination. In the case of the oscillation (frequency) mode, shear stress was applied in the regime of small-amplitude sinusoidal harmonic oscillations. The result of the measurements was the dependence of stress or strain on frequency. The data measured in the oscillation mode are analyzed using complex numbers. The study of mechanical properties in the oscillation mode allows characterizing them only qualitatively. Therefore, other modes are additionally used. The second method allows the construction of flow curves, on the basis of which the behavior of the prepared colloidal systems can be established. In case they behave like solids until the stress reaches a certain value called the yield point, elastic deformation is observed, while flow (irreversible deformation) begins after the transition through the yield point.

Frequency dependencies of storage and loss moduli and complex viscosity (Figure 5a–c) showed a transition from a solid-like to liquid-like state (marked by blue circles) with decreasing shear stress frequency values [41]. It is worth noting that the same behavior was observed in the compositions of suspensions in water after 7 days after NP synthesis. This additionally indicates the stability of nanosized PECs over time.

The fact of the formation of organized colloidal structures is also confirmed by the data of shear stress as a function of the shear rate (Figure 5d–f). The appearance of all curves corresponds to pseudoplastic liquids with defined yield stress. Its value practically does not change with increasing chitosan content in initial PEC compositions. The type of flow curves does not change significantly with TMZ immobilization. A significant difference was observed for the NP-01-05 composition (Figure 5f). Yield stress increased almost three times in comparison with the sample without drugs (from 0.22 Pa to 0.59 Pa). This may indicate certain coagulation processes that can subsequently lead to the aggregation of NPs containing higher chitosan content when immobilized with drugs. The work on rheological parameters of PEC suspensions in different nutrient media [42] is also of great interest from the point of view of studying the obtained colloidal systems under conditions as close as possible to medical practice. Such investigations are planned to be carried out by our research group in the near future.

### 3.5. Drug Immobilization and Cumulative Release of TMZ

The adsorption of TMZ onto polysaccharides’ matrix is determined by the pore size distribution and surface properties. The pore size of the NPs determines the number of drug molecules that can be adsorbed into it. The size of the NPs depends on the molecular weight of the chitosan and pectin polymer, as well as on the molar ratio of them. It is obvious that higher-molecular-weight initial polysaccharides produce larger structures.

Adsorption kinetics of the products was used to investigate the saturated adsorption capacity of pectin–chitosan–TMZ. The effect of the molar ratio on the adsorption of TMZ by NPs was studied at C_0_ = 50 mM and a ratio of pectin to chitosan of 0.1 wt.%:0.1 wt.% (NP-01-01) and 0.1 wt.%:0.5 wt.% (NP-01-05).

As shown in Figure 6a, the proportion of chitosan in the system increases and the immobilization efficiency of TMZ from the solution into NPs decreases. Perhaps this can be explained by the fact that the affinity of TMZ for pectin is greater than for chitosan.

The study of the dynamics of drug adsorption (Figure 6b) showed a strong interaction of the drug with the surface of the NPs at the initial stage of the loading process. This explains the sharp rise in the curves and the maximum speed of the process during the first 20 and 40 min for NPs of the first and second type, respectively.

Parameters from pseudo first-order [43] and pseudo second-order kinetic models [44] were fitted to the experimental data to evaluate the adsorption kinetics of TMZ uptake by pectin–chitosan NPs.

The linear form of the pseudo first-order model can be expressed as
$$log(q_{e}-q_{t})=logq_{e}-\frac{1}{2.303}k_{1}t$$
where $q_{e}$ and $q_{t}$, mM, are the adsorption capacities at equilibrium and at time t, respectively. The constants, $k_{1}$ and $q_{e}$, in the experiment were calculated using the slope and intercept of plots of $log(q_{e}-q_{t})$ versus t (Figure 7a, Table 3). Fit lines at both ratios yielded low $R^{2}$ values and so the application of the equation of the first order is inappropriate. This suggests that the adsorption of TMZ on pectin–chitosan NPs did not follow pseudo first-order kinetics.

The pseudo second-order rate expression can be linearly expressed as
$$\frac{t}{q_{t}}=\frac{1}{k_{2}q_{e}^{2}}+\frac{t}{q_{e}},$$
where $k_{2}$ (mM/min) is the rate constant for pseudo second-order adsorption and $k_{2}C_{e}^{2}$ (mM/min) is the initial adsorption rate.

The experimental values and the calculated value of parameters for the pseudo second-order models are shown in Figure 7b and Table 3. It is shown that the fitted equilibrium adsorption data at NP-01-01 are in close agreement with those observed experimentally, and the calculated correlation coefficient $R^{2}$ in this case is close to unity of pseudo second-order kinetics. Therefore, the TMZ adsorption by NP-01-01 can be described by the pseudo second-order kinetic model and the drug is adsorbed onto these NPs via a chemical interaction. This could probably explain the lower value of drug release from NP-01-01 particles into biological fluids, as will be shown below.

Various types of adsorption isotherm equations were used to describe the nature of the process. The adsorption of TMZ by pectin–chitosan NPs was modeled using the Freundlich and BET isotherms. An attempt was also made to assess the loading process by the Langmuir model. However, the nature of the location of the experimental data in the coordinates of the linear Langmuir equation showed the non-usability of the classical model for describing this experiment. The Langmuir model is based on the hypothesis of the formation of a monomolecular layer of adsorbate on the surface of the sorbent with equal energy and enthalpy of all active centers. It might be supposed that TMZ loading does not qualify for these conditions.

Then, it was shown that the experimental data have a linear dependence in the coordinates of the Freundlich model. The Freundlich isotherm is suitable for both monolayer and multilayer adsorption, and it assumes that the adsorbate is adsorbed on the heterogeneous surface of an adsorbent with different energy values, and the active sorption centers with maximum energy are filled first (Figure 8a). In the case of NP-01-05, a linear dependence is also observed in the coordinates of the BET model equation. For NP-01-01, this model apparently cannot be used to describe the sorption process ($R^{2}$ = 0.739) (Figure 8, Table 4).

The parameters of the Freundlich equation were found graphically (Table 4) and used to design the TMZ adsorption equation. It can be used to calculate the amount of the adsorbed drug from the solution under equilibrium conditions.

For the adsorption of TMZ by a sample of NP-01-01, the equation is as follows:$$a_{calc}=20.8\cdot {\left[C\right]}^{0.87}.$$

For sample NP-01-05,
$$a_{calc}=10.1\cdot {\left[C\right]}^{0.65}.$$

This suggested that the particles can suitably be an effective drug delivery vehicle. To confirm, experiments were conducted on the desorption of TMZ from pectin–chitosan particles into model solutions MBPF and MCF. As can be seen from the graph (Figure 9), the drug was released mainly during the first 30 min of the process. And so, the sample NP-01-01 had a greater loading value then NP-01-05 (Figure 6), but significantly lower release efficiency that varied between 42 and 45% for different media (Figure 9). This may be due to the presence of a chemical interaction between the drug and the sorbate, which is reflected in the kinetics of the process, as was shown above.

The maximum release rate was also observed during the first 30 min of the process and was slightly higher in the case of NP-01-05 (Figure 10). However, in general, productivity indicators, calculated at a relatively initial quantity of the drug, do not exceed 10% (Figure 11).

## 4. Discussion

Lyophilic colloidal systems are the most required structures for drug storage and delivery. They include microemulsions formed by surfactants, micelles, and particles of various sizes. Microemulsions have a number of advantages such as dependence of properties only on the composition and not on the mixing conditions of components, and a large internal volume of droplets. High stability and reproducibility of properties of both microemulsions and micelles are noted in the literature [45]. The main disadvantages include the presence of surface-active substances and co-surfactants in high concentrations. Materials for pharmaceutical applications should contain non-toxic and biocompatible components [46]. The creation of particles based on biopolymers or their derivatives is considered an excellent alternative from this point of view. Moreover, it is in the case of particles that it becomes possible to create colloidal nanoscale systems for exceptional cases such as diseases including malignant tumors of the brain [47]. In contrast to the above-mentioned emulsions and micelles, in the case of nano- and microparticles, the conditions and order of synthesis already play a major role along with the variation of composition. But this approach provides its advantages in the form of a wide range of properties, the fine-tuning of which will allow for solving many exclusive problems [48]. In the case of the brain, these include dimensions, surface charge, and mechanical or viscoelastic properties of the nano- and micro-materials themselves. The last parameter refers to the fact that more rigid NPs penetrate cells more effectively, compared to softer ones [49]. Also, an important factor determining the very fact of particle formation is the composition and ratio of initial components during their production.

Solutions for the synthesis of nano- and microparticles based on individual macromolecules with the addition of a cross-linking agent, which are common inorganic easily soluble salts, are abundant in the literature. It is worth noting the interest in polysaccharides, such as chitosan [50] and pectin [51], structures with simpler to use and more controlled molecules. Such work is characterized by the simplicity and speed of obtaining the final products, but also faces problems of reproducibility of the results. An alternative option for synthesizing colloidal systems is mixing solutions of oppositely charged polyelectrolytes. They enter into rapid electrostatic interactions, which leads to association. The associates are called polyelectrolyte complexes (PECs). Oppositely charged groups are neutralized during association, which causes the hydrophobization of the corresponding chain segments. Another feature is the cooperative nature of complexation, which is explained by the numerous charged functional groups in macromolecules. Due to their cooperativity and hydrophobicity, the associates have reduced solubility in media with different polarity. By tuning the initial mixing of polyelectrolyte solutions, the formation of micro- and NPs is also possible. In addition to good stability in the case of PECs, the above-mentioned fine-tuning of the physicochemical properties of the particles is necessary.

Many authors have demonstrated good biocompatibility and efficiency of drug immobilization with carriers based on pectin [52,53,54] and chitosan [55,56,57]. Drug delivery systems based on them have been tested in in vivo and in vitro models. In addition to excellent biocompatibility, pectin-based materials have the ability to slow down the proliferation of glioblastoma cells in contact with them [14], while chitosan-based materials show cytotoxic effects on cancer cells while having minimal toxicity on normal cells [58].

There are ionic and hydrogen interactions between NP carbohydrates, which could be chemically tuned by the different rate of acetylation, methylation, esterification, or de-esterification. These minor changes drastically modify both NPs’ intrinsic properties: shape, size, degradation velocity, and the rate of controlled drug release. Previously, we showed that the pectin esterification rate and Ca^2+^ concentration provide a significant effect on its viscoelastic properties, cell adhesion and division, and degradation in physiological conditions [22,59]. Chitosan chemical modifications provide the same effect, which could be used for NPs’ size adjustment [60], stability, and biodegradation rate [61]. The charge compositions of pectin and chitosan allow us to immobilize complex zwitterionic molecules for extended time to stabilize the drug concentration and provide the controlled release by the processes of nanocarrier biodegradation [62].

PECs based on chitosan and pectin have been studied for a long time, but mostly for the creation of implantable bulk hydrogels. Thus, in [63], such materials were studied for the immobilization and release of curcumin. The obtained constants of sorption kinetic models of the first and second order are comparable with the data obtained in our work, which confirms the reliability of our data.

Nanoscale PECs of chitosan and pectin were formed in [30] for the immobilization of taurine with respect to the treatment of gastrointestinal diseases. A relatively high immobilization efficiency of up to 60% and efficient drug yield was demonstrated. The particle sizes ranged from 42 to 74 nm and the zeta-potential ranged from +43.5 to +48.4 mV. In contrast to our results (the dimensions ranged from 30 to 300 nm according to TEM data and 58.54 to 252.3 nm according to AFM; zeta-potential ranged from −28.4 mV to 73.4 mV), the authors performed the experiments in an acidic environment close to the physiological environment. Under such conditions, the particles are indeed in a sufficiently stable state. In our case, the experimental pH should reach 7.4, which naturally affects the partial segregation and aggregation of unstable colloidal particles, especially with a predominant excess of chitosan. In addition, low-molecular-weight chitosan was used in the above work. The parameters of pectin are unfortunately not specified. The use of high-molecular-weight chitosan in our case is due to the possibility of using it to vary the mechanical properties of materials in a wide range of values [64].

Chitosan NPs were synthesized by the authors of [65]. The stabilization of the particles was achieved by their binding to polyurethane-based fibrils and doping with gold NPs. Their ability to absorb TMZ and the final effect on human GBM U87 cells were investigated. High drug immobilization efficiency was observed for single polysaccharide NPs compared to stabilized NPs. Tumor cell viability was significantly decreased when using drug delivery agents followed by stabilization compared to the original antitumor drug. It is worth noting that high-purity TMZ was used in the work, when a commercial drug was used in our work. Such a drug contains additional impurities in accordance with pharmacopoeia standards of production.

TMZ loading in the NPs was influenced by different factors, such as NPs’ surface charge, electrostatic attraction between TMZ and polysaccharides, and the difference ratio of pectin and chitosan in particles. These factors affected NP-01-01 to obtain a high load (Table 5). While NP-01-05 showed a lower TMZ immobilization than NP-01-01, it could point to the importance of pectin for drug loading in NPs.

In addition to polysaccharides, nanostructured lipid carriers can act as TMZ transporters [66,67]. Nanostructured lipid carriers have been developed for medical applications to deliver lipid-soluble drugs. Due to their structural similarity to biomembranes and safety, they are preferred over classical methods of drug delivery and treatment [68]. However, lipid carriers contain a solid lipid core and therefore encapsulate drugs, including TMZ, less effectively than polysaccharide NPs. Since TMZ is characterized by high lipophilicity, it can be expected that pectin–chitosan NPs would be a more suitable drug delivery method due to the expected higher solubility of the drug in the highly porous structure. It is known that TMZ contained in the structure of nanolipid carriers is more effective than that contained in NPs [69]. In addition, the positive charge of the amino group of chitosan increases the diffusion of the drug through the mucous membranes after intranasal delivery [13,70]. This means that it is possible to increase the accumulation of TMZ in the desired area and subsequently reduce the amount of drug administered and its cytotoxicity [71].

There is almost no information in the literature on the mechanical properties of individual NPs of PECs of chitosan with pectin. Therefore, the data on the mechanical properties of the NPs obtained in this work are rather difficult to evaluate fully. In our previous work [14], studies of pectin NPs were carried out and the present data are in good correlation with those data.

Based on the study of materials by IR spectroscopy of various types of microscopy, it can be assumed that NPs are gels with mesh cross-linked packing in our work. The formation of such a structure can be due to a large number of physical cross-links and bonds such as ionic, hydrogen bonds, including a sufficiently large number of charged amino and carboxyl groups identified by the above-mentioned method. It is such cross-linking that provides mechanical properties sufficient for absorption by tumor cells.

Our results presented in this article include studies of the mechanical properties of NPSs (Young’s modulus and adhesion) when one of the components in the initial suspension mixtures is changed and the content of the second component is unchanged, as well as diagrams of the influence of chitosan and pectin ratios on the mechanical properties. Thus, Young’s modulus ranged from 58 to 234 kPa, because of which the materials can be regarded as soft or as hard enough for cellular uptake. The adhesion of the NPs takes negative values from −0.3 to −3.57 pN. It may also mean less possibility of proteins from blood plasma to adhere on the surface of NPs.

NPs with a prevalence of chitosan in the stoichiometric polymer ratio have positive surface charge, which may make them suitable for drug delivery across the BBB. On average, the zeta-potential modulus decreases with increasing adhesion. Particles with 0.1 wt.% pectin concentration and different chitosan content have colloidal stability and are presumably able to cross the BBB.

Based on the studies presented above, it can be preliminarily concluded that the NP-01-01 sample (diameter value was 128.7 nm (IQR 87 nm) by AFM) is a preferable delivery agent for antitumor drugs to overcome the BBB when delivered to brain tumor cells. Nevertheless, the results obtained at this stage do not allow for a definite conclusion, as the use of different types of NPs has advantages and disadvantages compared to each other. Another issue is the number of NPs with specific drug concentrations required for the therapy. As has been shown by dynamic rheology, at certain particle amounts, there is a certain structurization of suspensions, possibly affecting the biological processes of the patient. Of course, there are various methods for the functionalization and stabilization of colloidal drug systems. So, in paper [72], a rather common method of PEGylation is described in detail. But even in this case, as the authors specify in their work, there are certain difficulties and disadvantages. In this regard, to overcome limitations to the use of NPs, it is necessary to control the stability of nanocarriers, target delivery systems to tumor cells, and optimize the amount of drug released. In addition, the application of multicomponent NPs can be challenging due to the number of parameters that need to be considered, such as dimensions, charge, purity, drug encapsulation efficiency, and conjugated ligand coating efficiency. Despite these challenges, efforts continue to optimize this approach, particularly the use of cellular receptor ligands such as transferrin receptor 1 (TfR1) to effectively target brain tumor cells [73].

Future research directions should include the ability of the drug delivery system based on PEC pectin and chitosan to cross the BBB, the mechanism of internalization with the tumor cell membrane, the efficiency of chemotherapy agent resorption, and the effect of suppressing tumor growth in in vitro and in vivo models.

Polysaccharide NPs have numerous potential applications in the clinic due to their biocompatibility, biodegradability, and versatility. The easy synthesis and drug loading make them a promising tool for personalized medicine by the drug addressing, concentration control, and excretion. The attachment of antibodies or nucleic acid aptamers to the NPs’ surface will drastically decrease drug side effects and reduce doses by nanocarrier focusing and concentration in the tumor focus [74]. The size variations and tunable chemical properties of PECs allow us to choose the best ones for brain-targeted delivery as the key prospect for our future work including in vitro and in vivo experiments on microfluidic microvessel systems and xenograft models.

## 5. Conclusions

The aim of this work was to develop a method for the formation of nanosized polyelectrolyte complexes based on oppositely charged macromolecules of chitosan and pectin polysaccharides. Special synthesis conditions were selected and the ratios of polyelectrolyte solutions were established, resulting in homogeneous suspensions of nanoparticles. Dependencies of surface charge by means of zeta-potential estimation, mechanical properties of nanoparticles and suspensions as a whole, and structure on the composition of initial mixtures were studied. A comprehensive evaluation of sorption properties and sorption kinetics of nanoparticles with respect to the antitumor drug temozolomide was performed. Among series of nanoparticles obtained, a polyelectrolyte vehicle based on an equal pectin–chitosan ratio (0.1%, *w*/*w*) was chosen as the most effective for overcoming the blood–brain barrier and temozolomide immobilization and delivery to brain tumors.

## Figures and Tables

**Figure 1 biomedicines-12-01393-f001:**
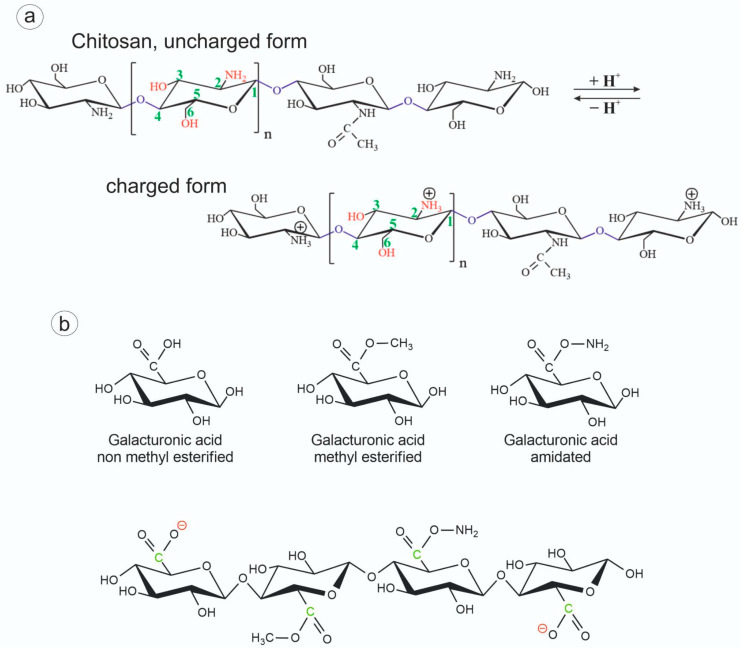
(**a**) The structural formula of the uncharged and charged forms of chitosan. The presented polysaccharide can contain both types of glucosamine residues connected by O-glycoside bonds simultaneously. (**b**) The general scheme of pectins’ structure: various modifications of galacturonic acid in pectins. The figure is based on [3].

**Figure 2 biomedicines-12-01393-f002:**
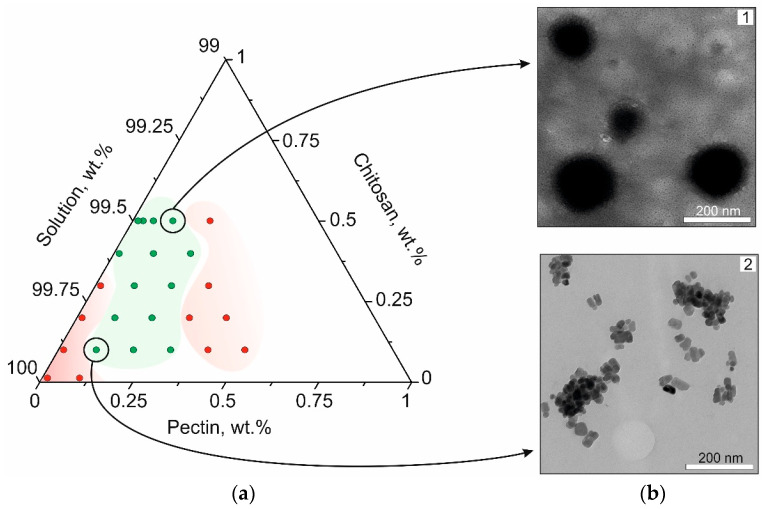
(**a**) A phase diagram of polyelectrolyte solutions. The red color indicates compositions for which there was no NP formation observed. The compositional region of the nanoscale PECs is highlighted in green. (**b**) High-resolution TEM images of initial PEC compositions: 1—chitosan at 0.5 wt.%, pectin at 0.1 wt.%; 2—chitosan at 0.1 wt.%, pectin at 0.1 wt.%.

**Figure 3 biomedicines-12-01393-f003:**
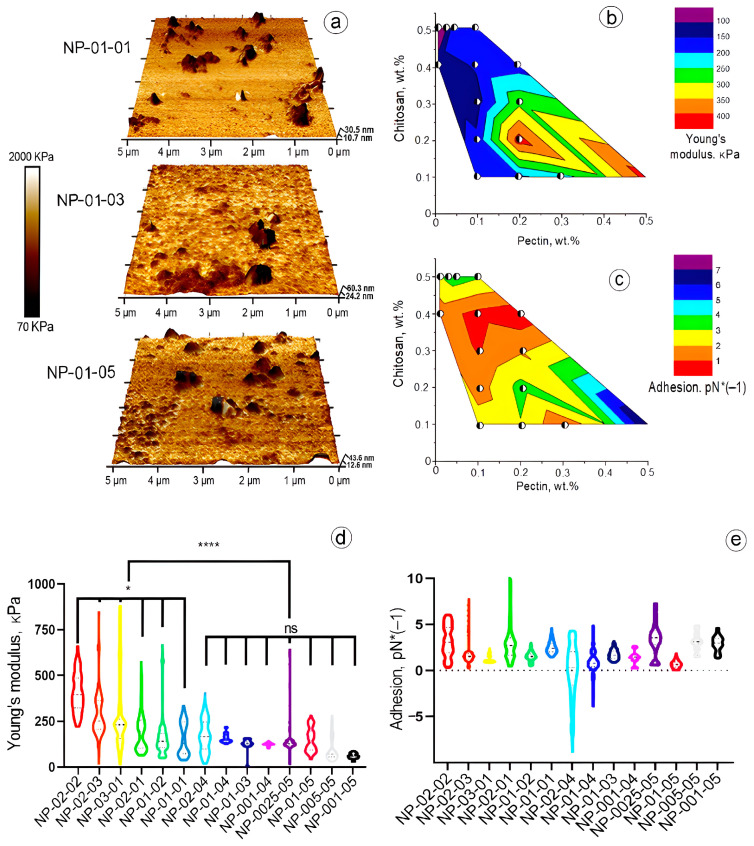
(**a**) Three-dimensional AFM images with the correlation of NPs’ sizes and Young’s modulus obtained with a mass ratio of pectin to chitosan at 0.1:0.1, 0.1:0.3, and 0.1:0.5; (**b**,**d**) type of Young’s modulus and (**c**,**e**) adhesion distributions of stechiometric ratio of pectin and chitosan of NPs. Statistical differences were designated as significant if the p-values were less than 0.05 (* *p* ≤ 0.05) or less than 0.0005 (**** *p* ≤ 0.0001). The Kruskal-Wallis test was used to assess the differences between two independent samples. All abbreviations of the obtained NP samples are presented below in Table 1.

**Figure 4 biomedicines-12-01393-f004:**
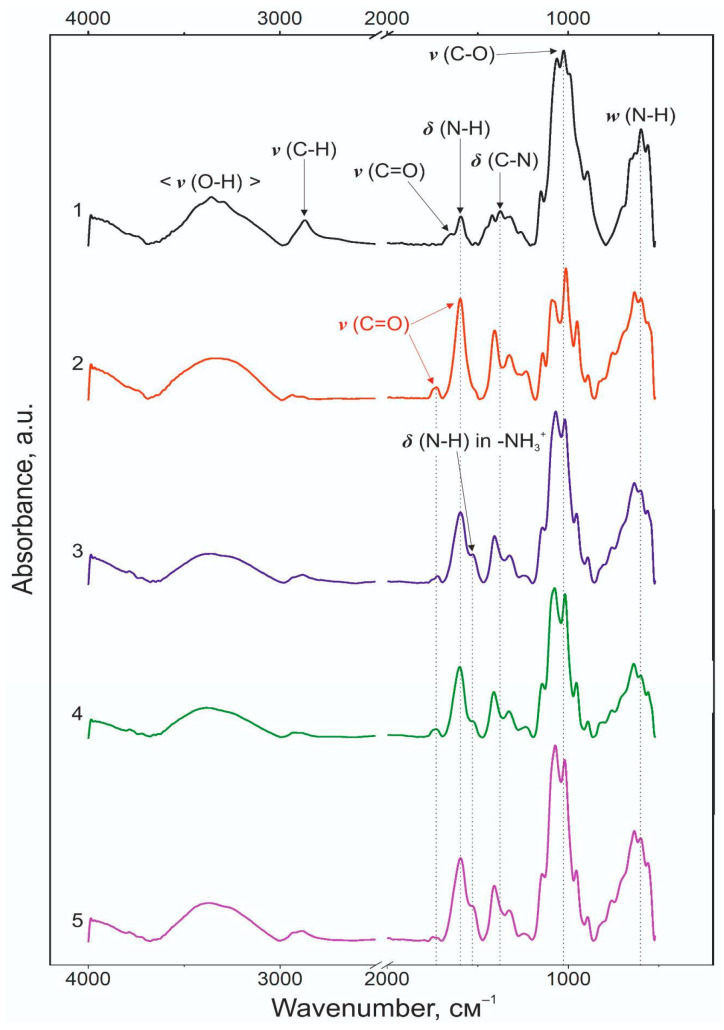
IR spectra of powders of (1) chitosan and (2) pectin, (3) NP-01-01 NPs, (4) sample with ratio of pectin to chitosan of 0.5 wt.%:0.1 wt.%, and (5) NP-01-05.

**Figure 5 biomedicines-12-01393-f005:**
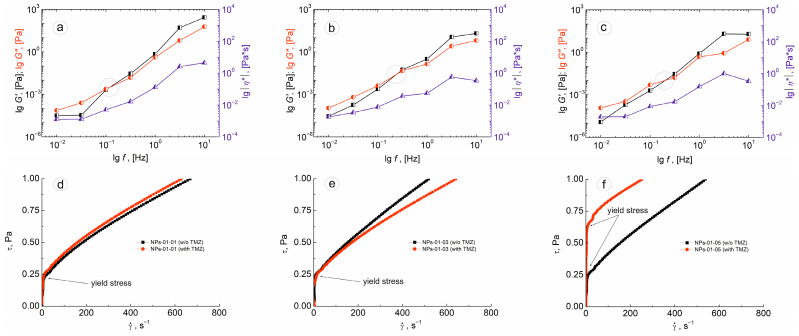
Plots of storage (black curves) and loss moduli (red curves) and complex viscosity (blue curves) as a function of shear stress frequency: (**a**) NPs-01-01, (**b**) NPs-01-03, (**c**) NPs-01-05. Point of transition region from a solid-like to liquid-like state marked by blue circles. Plots of shear stress versus the shear rate: (**d**) NPs-01-01, (**e**) NPs-01-03, (**f**) NPs-01-05. For plots d-f, black curves correspond to compositions without TMZ, and red curves correspond to PEC immobilized with drugs.

**Figure 6 biomedicines-12-01393-f006:**
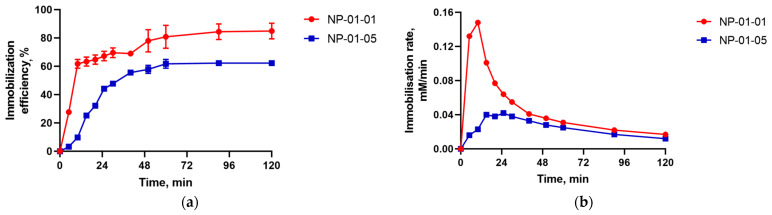
Effect of mass ratio of pectin–chitosan on adsorption kinetics of TMZ by NPs: (**a**) TMZ immobilization to pectin–chitosan particles; (**b**) TMZ immobilization rate to pectin–chitosan particles.

**Figure 7 biomedicines-12-01393-f007:**
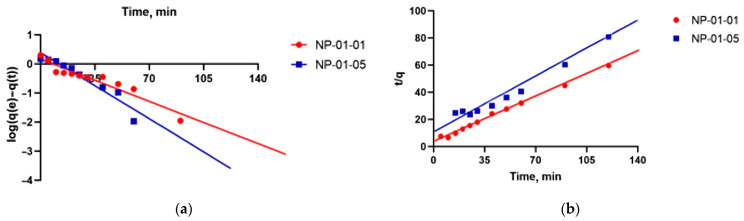
(**a**) Pseudo first-order kinetic and (**b**) pseudo second-order kinetic model fit for TMZ sorption onto pectin–chitosan NPs.

**Figure 8 biomedicines-12-01393-f008:**
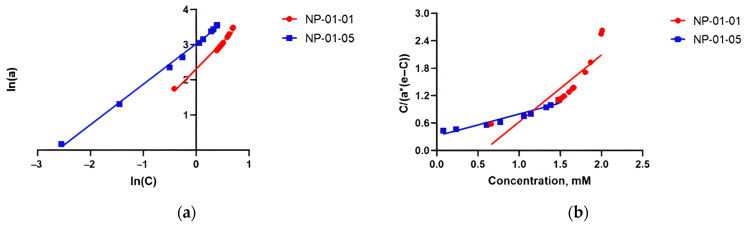
(**a**) Freundlich and (**b**) BET isotherms for TMZ adsorption on pectin–chitosan NPs.

**Figure 9 biomedicines-12-01393-f009:**
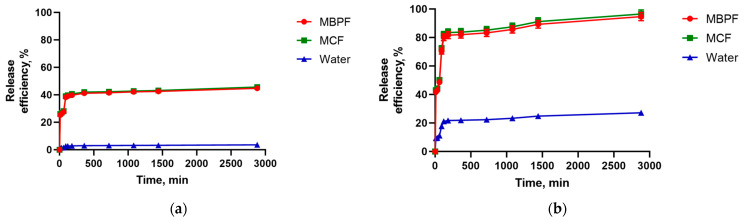
Cumulative release profile (%) of TMZ from pectin–chitosan NPs to different media at various ratios of pectin to chitosan: (**a**) NP-01-01; (**b**) NP-01-05.

**Figure 10 biomedicines-12-01393-f010:**
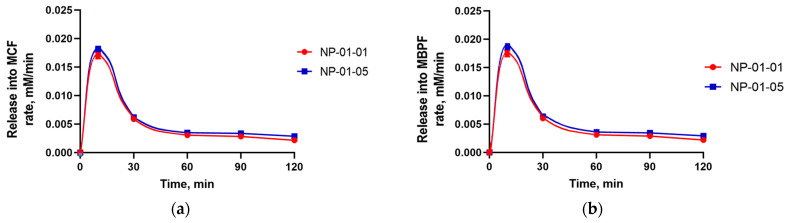
Release rate of TMZ from pectin–chitosan NPs at various ratios of pectin to chitosan: (**a**) into the MCF; (**b**) into the MBPF.

**Figure 11 biomedicines-12-01393-f011:**
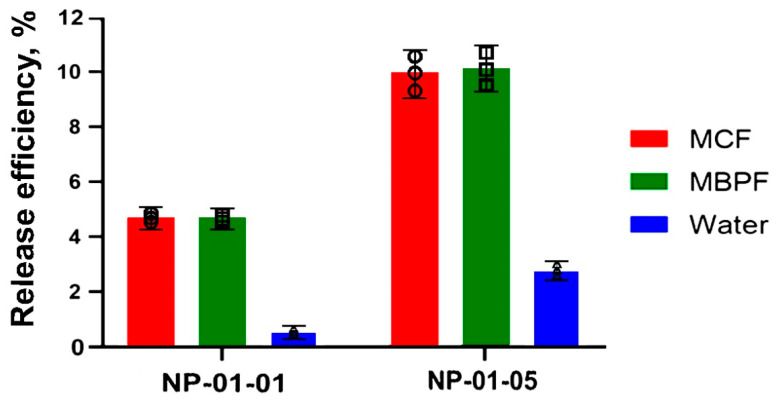
Overall efficiency of the pectin–chitosan–TMZ system. All data are presented as means and standard errors of the mean (M ± SEM). Circle mark—individual data points for MCF (in red). Square mark—individual data points for MBPF (in green). Triangle mark—individual data points for water (in blue).

**Table 1 biomedicines-12-01393-t001:** Compositions and physicochemical parameters of the obtained chitosan–pectin PEC samples.

Concentration Ratio of Pectin and Chitosan (wt. %)	Diameter of NPs Obtained by AFM, nm	Young’s Modulus, kPa	Adhesion, pN^−1^	Zeta-Potential, mV	PDI
0.3:0.1 (NP-03-01) *	230.2 (IQR 53.4)	84.82 (IQR 39.75)	1.15 (IQR 0.09)	−28.4	0.244
0.2:0.1 (NP-02-01)	194.3 (IQR 76.3)	131.90 (IQR 61.24)	2.73 (IQR 0.95)	−11.9	0.438
0.2:0.2 (NP-02-02)	394.5 (IQR 81.8)	8.82 (IQR 36.52)	3.10 (IQR 1.38)	−2.3	0.53
0.2:0.3 (NP-02-03)	252.3 (IQR 78.2)	84.13 (IQR 32.83)	1.52 (IQR 0.32)	47.1	0.462
0.2:0.4 (NP-02-04)	164.8 (IQR 73.7)	98.10 (IQR 38.44)	0.30 (IQR 1.82)	51.1	0.511
0.1:0.1 (NP-01-01)	128.7 (IQR 87)	145.40 (IQR 88.28)	2.42 (IQR 0.46)	23.2	0.182
0.1:0.2 (NP-01-02)	139.3 (IQR 38.4)	62.42 (IQR 23.14)	1.52 (IQR 0.28)	2.6	0.414
0.1:0.3 (NP-01-03)	129.0 (IQR 18.4)	62.09 (IQR 19.51)	1.68 (IQR 0.40)	10.8	0.479
0.1:0.4 (NP-01-04)	145.2 (IQR 16.6)	58.16 (IQR 18.36)	0.83 (IQR 0.48)	12.3	0.43
0.1:0.5 (NP-01-05)	130.1 (IQR 53.0)	87.82 (IQR 67.67)	0.66 (IQR 0.27)	73.4	0.435
0.05:0.5 (NP-005-05)	70.81 (IQR 49.8)	143.50 (IQR 82.18)	3.12 (IQR 0.73)	58	0.398
0.025:0.5 (NP-0025-05)	126.1 (IQR 11.7)	172.6 (IQR 80.3)	3.57 (IQR 0.80)	60	0.471
0.01:0.5 (NP-001-05)	58.54 (IQR 10.7)	164.80 (IQR 74.75)	2.95 (IQR 0.57)	55	0.351
0.01:0.4 (NP-001-04)	124.0 (IQR 3.0)	234.4 (IQR 143.53)	1.44 (IQR 0.28)	48.6	0.509

* Abbreviations of the obtained NP samples.

**Table 2 biomedicines-12-01393-t002:** FT-IR bands and their assignments. Correlation of lines was carried out on the basis of [37,38,39,40].

Functional Group, Vibration Modes, Classification	Chitosan, cm^−1^	Pectin, cm^−1^	NPs Chitosan at 0.1 wt.%–Pectin at 0.1 wt.%, cm^−1^	Sample Chitosan at 0.1 wt.%–Pectin at 0.5 wt.%, cm^−1^	NPs Chitosan at 0.5 wt.%–Pectin at 0.1 wt.%, cm^−1^
wagging vibrations of N-H and O-H	550 (m. *)	560 (m.)	561 (m.)	559 (m.)	560 (m.)
	589 (s.)	598 (s.)	599 (s.)	600 (s.)	600 (s.)
	641 (s.)	635 (s.)	636 (s.)	637 (s.)	635 (s.)
	696 (m.)	690 (m.)	697 (m.)	700 (m.)	700 (m.)
δ(C-H) corresponds to out-of-plane vibrations of glucosamine	899 (m.)	756 (m.)	759 (m.)	760 (m.)	760 (m.)
		807 (w.)	807 (w.)	810 (w.)	809 (w.)
		827 (w.)	827 (w.)	826 (w.)	830 (w.)
		892 (w.)	892 (w.)	892 (w.)	893 (w.)
ν(C-O-C) in glycosidic bond	-	-	953 (m.)	954 (m.)	955 (m.)
ν(C-O) asymmetric stretching in ring	1025 (v.s.)	1014 (v.s.)	1018 (v.s.)	1018 (v.s.)	1019 (v.s.)
ν(C-O-C) in pyranose ring structure	1068 (v.s.)	1089 (v.s.)	1068 (v.s.)	1075 (v.s.)	1070 (v.s.)
ν(C-O-C) in glycosidic bond	1150 (m.)	1142 (m.)	1142 (m.)	1143 (m.)	1142 (m.)
δ(O-H)	1251 (w.)	1233 (w.)	1245 (w.)	1239 (w.)	1243 (w.)
(1) δ(C-N)	1322 (m.)	1325 (m.)	1326 (m.)	1329 (m.)	1326 (m.)
(2) δ(C-H) ring stretching vibrations	1379 (m.)				
δ(C-H) corresponds to C-H group symmetrical deformation due	1422 (m.)	1407 (m.)	1406 (m.)	1411 (m.)	1408 (m.)
to the presence of saturated carbon atoms in the polysaccharide molecular structures					
δ(N-H) in ionized amino (-NH^3+^)	-	-	1527 (m.)	1524 (w.-m.)	1530 (m.)
(1) δ(N-H)	1591 (m.)	1597 (v.s.)	1597 (s.)	1600 (s.)	1594 (s.)
(2) ν_as_(C=O) in ionized carboxyl (-COO-) polysaccharides in salt form of pectins					
ν(C=O) in acetylated monosaccharide residues	1653 (w.)	-	-	-	-
ν(C=O) in COOH	-	1729 (w.)	1723 (w.)	1729 (w.)	1726 (w.)
			1750 (w.)	1750 (w.)	1750 (w.)
ν(C-H) in CH_2_ and CH_3_	2874 (w.)	2879 (w.)	2879 (w.)	2887 (w.)	2885 (w.)
		2936 (w.)	2932 (w.)	2929 (w.)	2931 (w.)
ν(N-H) in NH_3_	3292 (m.)	-	3285 (m.)	3275 (m.)	3276 (m.)
ν(O-H), water, and intramolecular H-bonds	3361 (m., br.), 3436 (m.)	3329 (m., br.)	3355 (m., br.)	3369 (m., br.)	3368 (m., br.)

* Intensities: w.—weak, m.—medium, s.—strong, v.s.—very strong, br.—broad.

**Table 3 biomedicines-12-01393-t003:** Adsorption kinetic model rate constants for TMZ adsorption on NPs.

Sample	$q_{e}$ (exp), mM	First-Order			Second-Order		
		$k_{1}$, min	$q_{e}$ (cal), mM	$R^{2}$	$k_{2}$, mol/(L*min)	$q_{e}$ (cal), mM	$R^{2}$
NP-01-01	2.0	0.0479	1.5	0.9116	0.0557	2.1	0.9964
NP-01-05	1.5	0.0748	2.5	0.9167	0.0267	1.8	0.9715

**Table 4 biomedicines-12-01393-t004:** Freundlich and BET isotherm model parameters and correlation coefficients for adsorption of TMZ on pectin–chitosan NPs.

Type	Calculated Parameters	NP-01-01	NP-01-05
Freundlich isotherm	$K_{F}$, (mmol/g)·(L/mmol)1/n	0.87	20.8
	1/n	0.65	10.1
	$R^{2}$	0.9793	0.9962
BET isotherm	$K_{BET}$, g/mmol	–0.75	2.57
	$a_{m}$, mmol/g	1.60	1.24
	$R^{2}$	0.739	0.9407

**Table 5 biomedicines-12-01393-t005:** TMZ immobilization profile. Data are represented as mean ± SD, n = 3.

Sample	TMZ Immobilization Efficiency, %	TMZ Immobilization Capacity, %
NP-01-01	84.1 ± 4.7	34.2 ± 3.6
NP-01-05	62.3 ± 2.7	24.8 ± 3.9

## Data Availability

Images and data are available from the corresponding author upon reasonable request. The data are not publicly available due to their originality.

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
