# Peer review of "Rational Design of Pectin–Chitosan Polyelectrolyte Nanoparticles for Enhanced Temozolomide Delivery in Brain Tumor Therapy"

_biomedicines, 2024, doi:10.3390/biomedicines12071393_

Round 1

Reviewer 1 Report

Comments and Suggestions for Authors

This study develops and characterizes pectin-chitosan polyelectrolyte nanoparticles designed for the delivery of temozolomide to brain tumors, optimizing their size, stability, and drug release properties. The findings highlight a 1:1 pectin-chitosan ratio as the most effective formulation, offering a promising approach for improving brain tumor therapy. The study holds significant importance and presents data well, but the manuscript requires major improvements to be suitable for publication. My comments are given below-

Title: The current title is informative but could be more concise and impactful. Consider changing it to: "Rational Design of Pectin-Chitosan Polyelectrolyte Nanoparticles for Enhanced Temozolomide Delivery in Brain Tumor Therapy."

Restructure Abstract: Problem Statement: Begin with a succinct statement of the biomedical challenge addressed: "Effective drug delivery systems for brain tumors remain a critical challenge in biomedical research."

Methods Overview: Briefly describe the innovative approach: "We developed polymeric nanoparticles through the rational design of polyelectrolyte complexes using oppositely charged pectin and chitosan."

Key Results: Highlight significant findings, such as nanoparticle size, zeta potential, and mechanical properties, and their relevance: "Nanoparticles exhibited diameters of 30-330 nm, zeta potentials from -29 to 73 mV, and Young's moduli from 58 to 422 KPa, indicating potential for effective drug delivery."

Conclusions: End with the main conclusion and its implications: "An equal mass ratio of pectin to chitosan (0.1%, w/w) was identified as the most effective for TMZ delivery, suggesting a promising approach for brain tumor therapy."

Explain the rheological measurements, including the type of rheometer used and the parameters (e.g., frequency, strain) assessed to understand the viscoelastic properties of the nanoparticles. Add catalogue number of all materials/ reagents and instruments model number with software used for the analysis. Cite https://doi.org/10.1021/acsomega.2c00472 with the sentence ´ Frequency dependencies of storage and loss moduli and complex viscosity (Fig. 5a-c) showed a transition from solid-like to liquid-like state (marked by blue circles) with decreasing shear stress frequency values ´

Results Analysis: Nanoparticle Size and Stability: Discuss the implications of the observed size range (30-330 nm) and zeta potential (-29 to 73 mV) for stability and cellular uptake. Relate these findings to optimal sizes for crossing the blood-brain barrier (BBB) and achieving sustained release. Nanomechanical Properties: Analyze the significance of the variations in Young's modulus (58 to 422 KPa) and adhesion forces (-0.72 to -3.53 pN). Discuss how these properties might affect nanoparticle behavior in the biological environment, including interactions with cells and the extracellular matrix. Mechanisms of Formation: Interpret the IR spectroscopy and rheology data to provide a mechanistic understanding of nanoparticle formation. Discuss the role of ionic interactions between carboxyl groups of pectin and amine groups of chitosan.

Drug Loading and Release Studies: Loading Efficiency: Describe the method used to quantify the amount of TMZ loaded into the nanoparticles. Discuss factors influencing loading efficiency, such as nanoparticle composition and preparation conditions. Release Kinetics: Provide detailed data on the release profiles of TMZ from the nanoparticles in model fluids at slightly alkaline pH. Discuss the release mechanism (e.g., diffusion, degradation) and compare it with desired release profiles for effective brain tumor therapy. Compare your nanoparticles with other existing drug delivery systems for TMZ, such as liposomes, micelles, or other polymeric nanoparticles. Highlight the advantages of your approach in terms of stability, drug loading capacity, release kinetics, and potential to overcome the BBB.

Discussion Depth: Expand the discussion to include potential clinical implications and future research directions. Discuss how your findings could translate into improved treatment outcomes for brain tumor patients and identify any limitations or challenges that need to be addressed in further studies.

References and Citations: Ensure all references are up-to-date and relevant to the field of drug delivery and nanoparticle research. Cite recent studies and reviews that support your methodology, findings, and conclusions. For example, with the sentence ´The use of high molecular weight chitosan in our case is due to the possibility of using it to vary the mechanical properties of materials in a large range of values´, cite https://doi.org/10.37349/ebmx.2024.00007 a recent report to make references up to date.

Mechanistic Insights and Potential for Clinical Translation: Provide a deeper mechanistic explanation of nanoparticle formation and drug release. Include discussions on how the specific interactions between pectin and chitosan influence the structural integrity and functionality of the nanoparticles. Discuss the potential for clinical translation of your nanoparticle system. Address scalability, reproducibility, and any regulatory considerations that might impact the development of these nanoparticles for clinical use. Include any preliminary in vivo data or future plans for animal studies.

Comments on the Quality of English Language

 Minor editing of English language required

Author Response

Dear reviewer, thank you for your questions. Answers to them are summarized below. Your comments have been considered when correcting the text of the article.

1) This study develops and characterizes pectin-chitosan polyelectrolyte nanoparticles designed for the delivery of temozolomide to brain tumors, optimizing their size, stability, and drug release properties. The findings highlight a 1:1 pectin-chitosan ratio as the most effective formulation, offering a promising approach for improving brain tumor therapy. The study holds significant importance and presents data well, but the manuscript requires major improvements to be suitable for publication. My comments are given below.

Title: The current title is informative but could be more concise and impactful. Consider changing it to: "Rational Design of Pectin-Chitosan Polyelectrolyte Nanoparticles for Enhanced Temozolomide Delivery in Brain Tumor Therapy."

1) Answer. We agree with the comments. Changes have been made in the Title.

2) Restructure Abstract: Problem Statement: Begin with a succinct statement of the biomedical challenge addressed: "Effective drug delivery systems for brain tumors remain a critical challenge in biomedical research." Methods Overview: Briefly describe the innovative approach: "We developed polymeric nanoparticles through the rational design of polyelectrolyte complexes using oppositely charged pectin and chitosan." Key Results: Highlight significant findings, such as nanoparticle size, zeta potential, and mechanical properties, and their relevance: "Nanoparticles exhibited diameters of 30-330 nm, zeta potentials from -29 to 73 mV, and Young's moduli from 58 to 422 KPa, indicating potential for effective drug delivery." Conclusions: End with the main conclusion and its implications: "An equal mass ratio of pectin to chitosan (0.1%, w/w) was identified as the most effective for TMZ delivery, suggesting a promising approach for brain tumor therapy."

2) Answer. The Abstract and Introduction sections have been revised and expanded.

3) Explain the rheological measurements, including the type of rheometer used and the parameters (e.g., frequency, strain) assessed to understand the viscoelastic properties of the nanoparticles. Add catalogue number of all materials/ reagents and instruments model number with software used for the analysis. Cite https://doi.org/10.1021/acsomega.2c00472 with the sentence ´ Frequency dependencies of storage and loss moduli and complex viscosity (Fig. 5a-c) showed a transition from solid-like to liquid-like state (marked by blue circles) with decreasing shear stress frequency values ´

3) Answer. The rheometer model, its principle of operation and main parameters are noted in Chapter 2.7 in sufficient details. Some clarifications have been introduced. Software used for the analysis: Chapter 2.12. Rheological measurement methods are described in detail in Chapter 3.4.

The proposed article uses another method called dynamic light scattering (DLS) microrheology. The authors of the article note that in comparison to conventional mechanical rheometry, DLS microrheology can measure much smaller sample volume and can measure over a much wider frequency range. It should be noted that modern mechanical rheometry allows to measure rather small volumes of materials and suspensions. In our case the sample volume for each measurement was 50 µl. A wide range of options and the possibility to vary the parameters very precisely make it possible to perform measurements in no way inferior to the method of dynamic light scattering (DLS) microrheology. And in some cases it allows us to obtain very interesting results. The sentence “Frequency dependencies of storage and loss moduli and complex viscosity (Fig. 5a-c) showed a transition from solid-like to liquid-like state (marked by blue circles) with decreasing shear stress frequency values” characterises well-known facts, unfortunately, not described in the article https://doi.org/10.1021/acsomega.2c00472, but, for example, found in the book [Mezger, Thomas. The Rheology Handbook: For users of rotational and oscillatory rheometers, Hannover, Germany: Vincentz Network, 2020. https://doi.org/10.1515/9783748603702].

4) Results Analysis: Nanoparticle Size and Stability: Discuss the implications of the observed size range (30-330 nm) and zeta potential (-29 to 73 mV) for stability and cellular uptake. Relate these findings to optimal sizes for crossing the blood-brain barrier (BBB) and achieving sustained release.

Nanomechanical Properties: Analyze the significance of the variations in Young's modulus (58 to 422 KPa) and adhesion forces (-0.72 to -3.53 pN). Discuss how these properties might affect nanoparticle behavior in the biological environment, including interactions with cells and the extracellular matrix.

Mechanisms of Formation: Interpret the IR spectroscopy and rheology data to provide a mechanistic understanding of nanoparticle formation. Discuss the role of ionic interactions between carboxyl groups of pectin and amine groups of chitosan.

Drug Loading and Release Studies: Loading Efficiency: Describe the method used to quantify the amount of TMZ loaded into the nanoparticles. Discuss factors influencing loading efficiency, such as nanoparticle composition and preparation conditions.

Release Kinetics: Provide detailed data on the release profiles of TMZ from the nanoparticles in model fluids at slightly alkaline pH. Discuss the release mechanism (e.g., diffusion, degradation) and compare it with desired release profiles for effective brain tumor therapy. Compare your nanoparticles with other existing drug delivery systems for TMZ, such as liposomes, micelles, or other polymeric nanoparticles. Highlight the advantages of your approach in terms of stability, drug loading capacity, release kinetics, and potential to overcome the BBB.

Discussion Depth: Expand the discussion to include potential clinical implications and future research directions. Discuss how your findings could translate into improved treatment outcomes for brain tumor patients and identify any limitations or challenges that need to be addressed in further studies.

4) Answer. The Discussion section has been revised and updated as recommended. The method for evaluation of TMZ loaded and TMZ loading efficiency was described. The methodology for the determination of temozolomide immobilized nanoparticles is expanded in Part 2 “Materials and Methods”. Detailed data about the TMZ release profile was added. To discuss the mechanism of the drug loading and unloading process in detail are needed the additional deeper investigations. This is planned for the future. Practical potential of nanoparticles, IR and rheology results has been described in Discussion section.

5) References and Citations: Ensure all references are up-to-date and relevant to the field of drug delivery and nanoparticle research. Cite recent studies and reviews that support your methodology, findings, and conclusions. For example, with the sentence ´The use of high molecular weight chitosan in our case is due to the possibility of using it to vary the mechanical properties of materials in a large range of values´, cite https://doi.org/10.37349/ebmx.2024.00007 a recent report to make references up to date.

5) Answer. The work with links has been done. In our opinion up-to-date article https://doi.org/10.1021/acssuschemeng.2c05843 is better suited for this sentence ´The use of high molecular weight chitosan in our case is due to the possibility of using it to vary the mechanical properties of materials in a large range of values´.

6) Mechanistic Insights and Potential for Clinical Translation: Provide a deeper mechanistic explanation of nanoparticle formation and drug release. Include discussions on how the specific interactions between pectin and chitosan influence the structural integrity and functionality of the nanoparticles. Discuss the potential for clinical translation of your nanoparticle system. Address scalability, reproducibility, and any regulatory considerations that might impact the development of these nanoparticles for clinical use. Include any preliminary in vivo data or future plans for animal studies.

6) Answer. We revised the discussion in order to your recommendations and emphasise the role of nanoparticle mechanical and chemical properties, including the role of pectin and chitosan as polyelectrolyte components (lines 660-673). We also discussed the nanoparticle system translation and scalability and provided the future plans (lines 815-824).

Reviewer 2 Report

Comments and Suggestions for Authors

1. It is better to introduce the charge density of chitosan and pectin polysaccharides. Please show the molecular structure in the paper. Please draw a cartoon image to show the PEC NPs.

2. Is the stability good in the presence of salt? PEC is largely affected by salt.

3. Crosslinking may be one method to enhance the stability of PEC. The authors can discuss (https://doi.org/10.1002/anie.202004180).

4. Some results need to be discussed in detail. For example, what is trend of mechanical properties change and the related reason?

5. "Among 26 series of NPs obtained polyelectrolyte vehicle based on equal pectin-chitosan ratio (0.1%, w/w) was chosen as most effective for TMZ delivery to brain tumors." What do you mean it is effective for brain tumor? In fact, the authors did not show any therapeutic results. I also do not know how this PEC can cross BBB.

6. Usually, PEC without PEG is unstable and has no stealth property. The authors at least should discuss this issue (https://doi.org/10.1016/j.addr.2023.114895).

Author Response

Dear reviewer, thank you for your questions. Answers to them are summarized below. Your comments have been considered when correcting the text of the article.

  1. 1) It is better to introduce the charge density of chitosan and pectin polysaccharides. 2) Please show the molecular structure in the paper. 3) Please draw a cartoon image to show the PEC NPs.
  2. Answer.
  • Zeta-potential data of initial solutions has been introduced (line 375).
  • Figure containing polysaccharides molecular structures has been added (line 123).
  • Despite the large amount of data obtained, it is necessary to continue research in this direction for the final description of the formation mechanism of polyelectrolyte complexes of chitosan and pectin according to our proposed method. We can already state that the obtained structures represent a complicated structure mesh polymer matrix, where macromolecules are connected both by physical entanglements and by a large number of different bonds, including electrostatic, hydrogen and others. The PEC particles can be graphically represented only by highly simplifying their real structure, so they were not indicated in this paper.
  1. Is the stability good in the presence of salt? PEC is largely affected by salt.
  2. Answer.

Yes, the addition of salt solutions at certain parameters of polyelectrolyte complexes leads to segregation and further aggregation. In the presented work, the stability of nanoparticles was carried out in model blood plasma fluid. A description of the method and brief result are given in parts 2.10 (line 278) and 3.1 (line 361). “The studies were performed by AFM for three days daily .... It is noted that most of the NPs maintained their stability. A slight segregation was observed for samples with high chitosan content in the initial mixtures.” Nanoparticle suspensions without AFM control visually maintained their stability after two weeks, although the experiments were conducted with freshly prepared samples. It is planned to continue the stability experiment and conduct longer study experiments under different conditions and in solutions of different compositions in the future.

  1. Crosslinking may be one method to enhance the stability of PEC. The authors can discuss (https://doi.org/10.1002/anie.202004180).
  2. Answer.

Thank you for presenting this interesting work. Yes, intermolecular crosslinking plays an important role in the formation and stabilization of polymer based nano- and microparticles (There is discussion from line 712). In the case of particles consisting of a single polymer or multiple polymers but not interacting with each other and forming physical gels it becomes necessary to use the conterions as crosslinking agents. The alternative is complexation or crosslinking of solutions of oppositely charged polymers. In the proposed work (https://doi.org/10.1002/anie.202004180), one of aspatamide derivatives as a polycation and PEG derivative as a polyanion were used. In fact, the authors of this work obtained the same polyelectrolyte complex as in our work. As well as in the proposed work, it is the maximum compensation of opposite charges that is the driving force for self-assembly with the formation of nanoscale spherical particles and, most importantly, stable for a long time. The difference with the proposed work is that our task was to obtain nanoparticles based on carbohydrates of natural origin. This is a more difficult task compared to synthetic polymers.

  1. Some results need to be discussed in detail. For example, what is trend of mechanical properties change and the related reason?
  2. Answer.

We emphasise the role of nanoparticle mechanical and chemical properties, including the role of pectin and chitosan as polyelectrolyte components (lines 686-699).

  1. "Among 26 series of NPs obtained polyelectrolyte vehicle based on equal pectin-chitosan ratio (0.1%, w/w) was chosen as most effective for TMZ delivery to brain tumors." What do you mean it is effective for brain tumor? In fact, the authors did not show any therapeutic results. I also do not know how this PEC can cross BBB.
  2. Answer.

Supplementary information has been added to the introduction (from line 79) and discussion of results (from line 700).

  1. Usually, PEC without PEG is unstable and has no stealth property. The authors at least should discuss this issue (https://doi.org/10.1016/j.addr.2023.114895).
  2. Answer.

The paper https://doi.org/10.1016/j.addr.2023.114895 describes in detail a rather effective and frequently used in laboratory and industrial practice method of colloid systems stabilization - PEGylation. The method involves coating nano- and microparticles with a layer (or layers) of nonionic polymer polyethylene glycol. The efficiency and simplicity of this approach, especially in terms of using an available water-soluble substance, is quite unquestionable. This approach was not considered in our work for several reasons.

First, the goal of our work was to create nanoscale particles with a controlled size, where the advantage will be given to a diameter of ~ 200 nm, and predominantly positive surface charge. Such limitations are based on preliminary data from the literature on the subject of the work. Preliminary - because the issue of overcoming the blood-brain barrier and the arrival of drugs and drug delivery agents in the brain is still relevant and not fully resolved. As noted above, at this stage of work stabilization of nanoparticles was achieved by varying the ratio of initial components and achieving maximum crosslinking. At the same time, as shown in our work, both the size and surface charge were varied. Additional introduction of stabilizing agents, such as PEG, will probably lead to a significant increase in stability and achieve the “stealth” effect, but will also contribute to an increase in particle diameter and change in surface charge. This may also tell on further surface modification of our obtained nanostructures with antibodies towards brain tumour cells and fluorescent tags (these types of works have already started).

Secondly, we can quote part of the text of the article https://doi.org/10.1016/j.addr.2023.114895 from page 7. “In turn, stealth nanomaterials/DDS may eventually become less biocompatible after metabolism in the body and trigger adverse effects. In addition, anti-fouling properties may or may not lead to a stealth effect and vice versa. A causal relationship between them is still highly context dependent. Such inference is well supported by our analysis below. Most of the PEGylated nanomaterials do not show a stealth effect while it goes without saying that PEGylation would decrease biofouling. In fact, biofouling of nanomaterials such as complement activation and opsonization may accelerate blood clearance. Alternatively, some types of biofouling such as albumin adsorption and hitchhiking onto the circulatory cells may render nanomaterials less visible to RES. It is worth recalling again that the standard method for judgement of stealth effect is via in vivo blood circulation profile at low doses. In vitro experiments such as protein adsorption and cellular uptake can only serve as indirect complementary evidence of the stealth effect.”.

To summarize, we can say that improvement of stability and interfacial properties of colloidal particles based on chitosan and pectin is possible, including the popular method of PEGgelation, but this is a separate big question, which will certainly be solved in the framework of the continuation of the project.

Round 2

Reviewer 1 Report

Comments and Suggestions for Authors

accept

Author Response

Dear reviewer, thank you again for your feedback and comments. 

Reviewer 2 Report

Comments and Suggestions for Authors

The PDI of nanoparticle is really large, indicating very broad distribution. Please double check it and consider this issue, giving some discussion.

Author Response

The PDI of nanoparticle is really large, indicating very broad distribution. Please double check it and consider this issue, giving some discussion.

Answer.

Dear reviewer, Thank you again for your comment.

We have critically analyzed the results obtained. Recalculation of some PDI values and additional measurements for some of the samples resulted in data that did not exceed the value of 0.7, which is suitable for the dynamic light scattering method [Tantra R. Nanomaterials Characterisation: An Introduction. Hoboken, New Jersey: John Wiley & Sons, Inc., 2016. doi: 10.1002/9781118753460]. The reason for the previous high values may have been insufficient sample preparation.

Your comments have been taken into account when correcting the text of the article. The values have been corrected in Table 1. Additional description has been introduced in the text (lines 353-359).
